# NeRF-SOS: Any-View Self-supervised Object Segmentation on Complex Scenes

**Zhiwen Fan[1], Peihao Wang[1], Yifan Jiang[1], Xinyu Gong[1], Dejia Xu[1], Zhangyang Wang[1]**
[1]Department of Electrical and Computer Engineering, University of Texas at Austin
`{zhiwenfan,atlaswang}@utexas.edu`

## Abstract

Neural volumetric representations have shown the potential that Multi-layer Perceptrons (MLPs) can be optimized with multi-view calibrated images to represent scene geometry and appearance without explicit 3D supervision. Object segmentation can enrich many downstream applications based on the learned radiance field. However, introducing hand-crafted segmentation to define regions of interest in a complex real-world scene is non-trivial and expensive as it acquires per view annotation. This paper carries out the exploration of self-supervised learning for object segmentation using NeRF for complex real-world scenes. Our framework, called *NeRF with Self-supervised Object Segmentation* (**NeRF-SOS**), couples object segmentation and neural radiance field to segment objects in any view within a scene. By proposing a novel collaborative contrastive loss in both appearance and geometry levels, NeRF-SOS encourages NeRF models to distill compact geometry-aware segmentation clusters from their density fields and the self-supervised pre-trained 2D visual features. The self-supervised object segmentation framework can be applied to various NeRF models that both lead to photo-realistic rendering results and convincing segmentation maps for both indoor and outdoor scenarios. Extensive results on the *LLFF*, *BlendedMVS*, *CO3Dv2*, and *Tank & Temples* datasets validate the effectiveness of NeRF-SOS. It consistently surpasses 2D-based self-supervised baselines and predicts finer object masks than existing supervised counterparts. Code is available at: `https://github.com/VITA-Group/NeRF-SOS`.

## 1 Introduction

Scene modeling and representation are essential to the computer vision community. For instance, portable Augmented Reality (AR) devices such as the Magic Leap One can reconstruct the scene geometry and localize users (DeChicchis, 2020). but they often struggle to comprehend the surrounding objects. This limitation poses challenges when designing interactions between humans and the environment. Although human-annotated data from diverse environments could mitigate the hurdles of understanding and segmenting the surrounding objects, collecting such data is often costly and time-consuming. Therefore, there is growing interest in developing intelligent geometry modeling frameworks that can learn from unsupervised or self-supervised techniques.

Recently, neural volumetric rendering techniques, such as neural radiance field (NeRF) and its variants (Mildenhall et al., 2020a; Zhang et al., 2020; Barron et al., 2021), have demonstrated exceptional performance in scene reconstruction, utilizing multi-layer perceptrons (MLPs) and calibrated multi-view images to generate fine-grained, unseen views. While several recent works have explored scene understanding with these techniques (Vora et al., 2021; Yang et al., 2021; Zhi et al., 2021), they often require either dense view annotations to train a heavy 3D backbone for capturing semantic representations (Vora et al., 2021; Yang et al., 2021), or human intervention to provide sparse semantic labels (Zhi et al., 2021). Although recent self-supervised object discovery approaches on neural radiance fields (Yu et al., 2021c; Stelzner et al., 2021) have been effective in decomposing objects on synthetic indoor data, there is still a significant gap to be filled in applying these approaches to complex real-world scenarios.

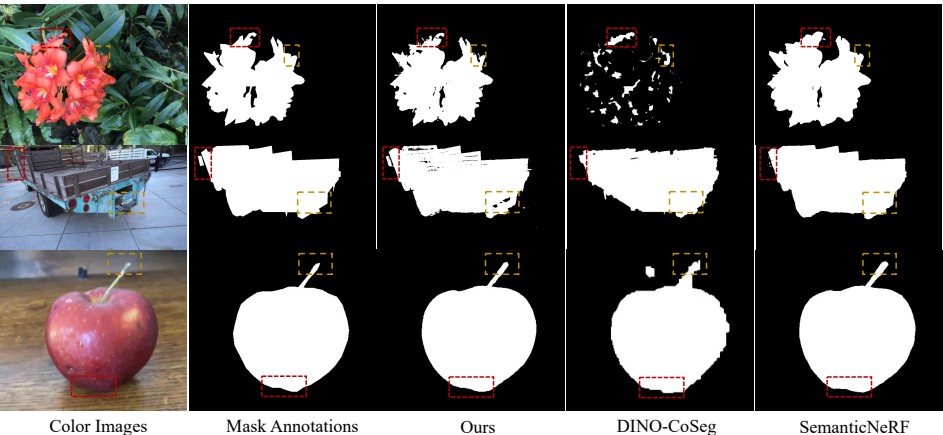

| Color Images | Mask Annotations | Ours | DINO-CoSeg | SemanticNeRF |

Figure 1: **Visual examples**. From left to right: ground truth color images, annotated object masks, object masks rendered by NeRF-SOS, 2D image co-segmentation using DINO (Amir et al., 2021), and object masks rendered by Semantic-NeRF (Zhi et al., 2021), respectively. NeRF-SOS outperforms the previous methods by generating object masks with more precise local details.

In contrast to previous works, we investigate a more generic setting, by using general NeRF models to segment 3D objects in real-world scenes. We propose a new self-supervised object segmentation framework for NeRF that utilizes a collaborative contrastive loss. Our approach combines features from a self-supervised pre-trained 2D backbone ("appearance level") with knowledge distilled from the geometry cues of a scene, using the density field of NeRF representations ("geometry level"). To be more specific, we adopt a self-supervised approach to learn from a pre-trained 2D feature extractor, such as DINO-ViT (Caron et al., 2021) and incorporate the inter-view visual correlations to generate distinct segmentation feature clusters within the NeRF framework. We introduce a geometry-level contrastive loss by formulating a geometric correlation volume between NeRF's density field and the segmentation clusters to make the learned feature clusters aware of scene geometry. Our proposed self-supervised object segmentation framework tailored for NeRF, dubbed **NeRF-SOS**, serves as a general implicit framework and can be applied to any existing NeRF models with end-to-end training. We implement and evaluate NeRF-SOS, using vanilla NeRF (Mildenhall et al., 2020a) for real-world forward-facing datasets (LLFF (Mildenhall et al., 2019)), object-centric datasets (BlendedMVS (Yao et al., 2020) and CO3Dv2 (Reizenstein et al., 2021)); and using NeRF++ (Zhang et al., 2020) for outdoor unbounded dataset (Tank and Temples (Riegler & Koltun, 2020)). Experiments show that NeRF-SOS significantly outperforms existing object discovery methods and produces view-consistent segmentation clusters: a few examples are shown in Figure 1.

We summarize the main contributions as follows:

- We explore how to effectively apply the self-supervised learned 2D visual feature for 3D representations through an appearance contrastive loss, which forms compact feature clusters to allow any-view object segmentation in complex real-world scenes.

- We propose a new geometry contrastive loss for object segmentation. By leveraging its density field, our proposed framework injects scene geometry into the segmentation field, making the learned segmentation clusters geometry-aware.

- The proposed collaborative contrastive framework can be implemented upon NeRF and NeRF++, for object-centric, indoor, and unbounded real-world scenarios. Experiments show that our self-supervised object segmentation quality consistently surpasses 2D object discovery methods and even yields finer segmentation results than the supervised NeRF counterpart (Zhi et al., 2021).

## 2 RELATED WORK

**Neural Radiance Fields**  NeRF is first proposed by Mildenhall *et al.* (Mildenhall et al., 2020b), which models the underlying 3D scenes as continuous volumetric fields of color and density via layers of MLP. The input of a NeRF is a 5D vector, containing a 3D location $(x, y, z)$ and a 2D viewing direction $(\theta, \phi)$. Several following works emerge trying to address its limitations and improve

the performance, such as unbounded scenes training (Zhang et al., 2020; Barron et al., 2021), fast training (Sun et al., 2021; Deng et al., 2021), efficient inference (Rebain et al., 2020; Liu et al., 2020; Lindell et al., 2020; Garbin et al., 2021; Reiser et al., 2021; Yu et al., 2021a; Lombardi et al., 2021), better generalization (Schwarz et al., 2020a; Trevithick & Yang, 2020; Wang et al., 2021b; Chan et al., 2020; Yu et al., 2021b; Johari et al., 2021; Varma T et al., 2022), supporting unconstrained scene (Martin-Brualla et al., 2020; Chen et al., 2021; Xu et al., 2022), editing (Liu et al., 2021; Jiakai et al., 2021; Wang et al., 2021a; Jang & Agapito, 2021; Kundu et al., 2022; Fan et al., 2022), multi-task learning (Zhi et al., 2021). In this paper, we treat NeRF as a powerful implicit scene representation and study how to segment objects from a complex real-world scene without any supervision.

**Object Co-segmentation without Explicit Learning**    Our work aims to discover and segment visually similar objects in the radiance field and render novel views with object masks. It is close to the object co-segmentation (Rother et al., 2006) which aims to segment the common objects from a set of images (Li et al., 2018). Object co-segmentation has been widely adopted in computer vision and computer graphics applications, including browsing in photo collections (Rother et al., 2006), 3D reconstruction (Kowdle et al., 2010), semantic segmentation (Shen et al., 2017), interactive image segmentation (Rother et al., 2006), object-based image retrieval (Vicente et al., 2011), and video object tracking/segmentation (Rother et al., 2006). The authors in (Rother et al., 2006) first shows that segmenting two images outperforms the independent counterpart. This idea is analogous to the contrastive learning way in later approaches. Especially, the authors in (Hénaff et al., 2022) propose the self-supervised segmentation framework using object discovery networks. (Siméoni et al., 2021) localizes the objects with a self-supervised transformer. The paper (Hamilton et al., 2022) introduces the feature correspondences that distinguish between different classes. Most recently, a new co-segmentation framework based on DINO feature (Amir et al., 2021) has been proposed and achieves better results on object co-segmentation and part co-segmentation.

However, extending 2D object discovery to NeRF is non-trivial as they cannot learn the geometric cues in multi-view images. uORF (Yu et al., 2021c) and ObSuRF (Stelzner et al., 2021) use slot-based CNN encoders and object-centric latent codes for unsupervised 3D scene decomposition. COLF (Smith et al., 2022) proposes a light field compositor module to accelerate NeRF-based object decomposition. Although they enable unsupervised 3D scene segmentation and novel view synthesis, experiments are on synthetic datasets with pre-defined categories, leaving a gap for complex real-world applications. NVOS (Ren et al., 2022) leverages users' scribbles for weakly-supervised object segmentation. The concurrent work, RFP (Liu et al., 2022) enables label-free object segmentation in real-world scenes with a propagation strategy. Panoptic NeRF Field (Kundu et al., 2022) proposes to represent each object instance using a separate MLP with supervisions from other models. N3F (Tschernezki et al., 2022) minimizes the distance between NeRF's rendered feature and 2D feature for scene editing. Most recently, DFFs (Kobayashi et al., 2022) propose to distill the visual feature from supervised CLIP-LSeg or self-supervised DINO into a 3D feature field via an element-wise feature distance loss function. It can discover the object using a query text prompt or a patch. In contrast, we design a new collaborative contrastive loss on both appearance and geometry levels to find the objects with a similar appearance and location without any annotations. The collaborative design is general and can be plug-and-play to different NeRF models.

## 3    METHOD

**Overview**    This paper presents an extension to existing NeRF models to enable object segmentation. As shown in Figure 2, we augment NeRF models by appending a parallel segmentation branch to predict point-wise implicit segmentation features. Specifically, NeRF-SOS can render depth ($\sigma$), segmentation ($s$), and color ($c$). We then use a self-supervised pre-trained framework (such as DINO-ViT (Caron et al., 2021)) to generate a feature tensor ($f$) from the rendered color patch ($c$), constructing an appearance-segmentation correlation volume between $f$ and $s$. Similarly, we instantiate a geometry-segmentation correlation volume using $\sigma$ and $s$. By generating positive/negative pairs from different views, we can distill the correlation patterns in both the visual feature and scene geometry into the compact segmentation field $s$. During inference, we use a clustering operation (such as K-means) to generate object masks based on the rendered feature field.

### 3.1    PRELIMINARIES

**Neural Radiance Fields**    NeRF (Mildenhall et al., 2020a) represents 3D scenes as radiance fields via several layer MLPs, where each point has a value of color and density. Such a radiance field can

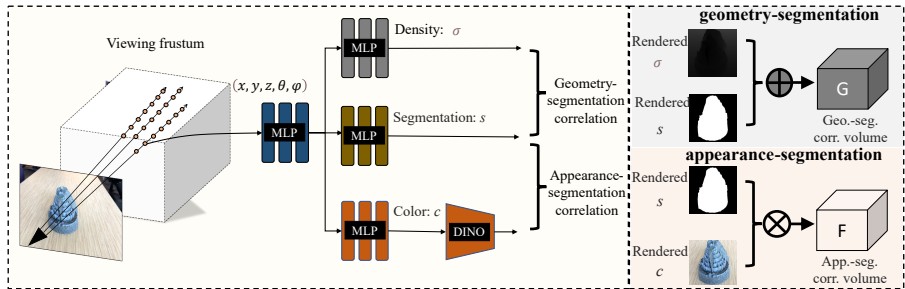

Figure 2: **The overall pipeline of the proposed NeRF-SOS.** Input with rays cast from multiple views, we render the corresponding color patch ($c$), segmentation patch ($s$), and depth patch ($\sigma$). Then, appearance-segmentation correlations and geometry-segmentation correlations are used to formulate a collaborative contrastive loss, enabling NeRF-SOS to render object masks from any viewpoint using the distilled segmentation field.

be formulated as $F : (\boldsymbol{x}, \boldsymbol{\theta}) \mapsto (\boldsymbol{c}, \sigma)$, where $\boldsymbol{x} \in \mathbb{R}^3$ is the spatial coordinate, $\boldsymbol{\theta} \in [-\pi, \pi]^2$ denotes the viewing direction, and $\boldsymbol{c} \in \mathbb{R}^3, \sigma \in \mathbb{R}_+$ represent the RGB color and density, respectively. To form an image, NeRF traces a ray $\boldsymbol{r} = (\boldsymbol{o}, \boldsymbol{d}, \boldsymbol{\theta})$ for each pixel on the image plane, where $\boldsymbol{o} \in \mathbb{R}^3$ denotes the position of the camera, $\boldsymbol{d} \in \mathbb{R}^3$ is the direction of the ray, and $\boldsymbol{\theta} \in [-\pi, \pi]^2$ is the angular viewing direction. Afterwards, NeRF evenly samples $K$ points $\{t_i\}_{i=1}^K$ between the near-far bound $[t_n, t_f]$ along the ray. Then, NeRF adopts volumetric rendering and numerically evaluates the ray integration (Max, 1995) by the quadrature rule:

$$\boldsymbol{C}(\boldsymbol{r}) = \sum_{k=1}^K T(k)(1 - \exp(-\sigma_k \delta_k))\boldsymbol{c}_k \quad \text{where } T(k) = \exp\left(-\sum_{l=1}^{k-1} \sigma_l \delta_l\right), \tag{1}$$

where $\delta_k = t_{k+1} - t_k$ are intervals between sampled points, and $(\boldsymbol{c}_k, \sigma_k) = F(\boldsymbol{o} + t_k\boldsymbol{d}, \boldsymbol{\theta})$ are output from the neural network. With this forward model, NeRF optimizes the photometric loss between rendered ray colors and ground-truth pixel colors defined as follows: $\mathcal{L}_{photometric} = \sum_{(\boldsymbol{r}, \widehat{C}) \in \mathcal{R}} \left\| \boldsymbol{C}(\boldsymbol{r}) - \widehat{C} \right\|_2^2$ where $\mathcal{R}$ defines a dataset collecting all pairs of ray and ground-truth colors from captured images.

## 3.2 Cross View Appearance Correspondence

**Semantic Correspondence across Views** Tremendous works have explored and demonstrated the importance of object appearance when generating compact feature correspondence across views (Hénaff et al., 2022; Li et al., 2018). This peculiarity is then utilized in self-supervised 2D semantic segmentation frameworks (Hénaff et al., 2022; Li et al., 2018; Chen et al., 2020) to generate semantic representations by selecting positive and negative pairs with either random or KNN-based rules (Hamilton et al., 2022). Drawing inspiration from these prior arts, we construct the visual feature correspondence for NeRF at the appearance using a heuristic rule. To be more specific, we leverage the self-supervised model (e.g., DINO-ViT (Caron et al., 2021)) learned from 2D image sets to distill the rich representations into compact and distinct segmentation clusters. A four-layer MLP is appended to segment objects in the radiance field parallel to the density and appearance branches. During training, we first render multiple image patches from different viewpoints using Equation 1, then we feed each batch into DINO-ViT to generate feature tensors of shape $H' \times W' \times C'$. They are then used to generate the appearance correspondence volume (Teed & Deng, 2020; Hamilton et al., 2022) across views, measuring the similarity between two regions of different views:

$$F_{hwh'w'} = \sum_c \frac{f_{chw}}{|f_{hw}|} \frac{f'_{ch'w'}}{|f'_{h'w'}|}, \tag{2}$$

where $f$ and $f'$ stand for the extracted DINO feature from two random patches in different views, $c$ is the feature dimension of DINO, $(h, w)$ and $(h', w')$ denote the spatial information on feature tensor for $f$ and $f'$, respectively, and the $c$ traverses through the feature channel dimension.

**Distilling Semantic Correspondence into Segmentation Field** The correspondence volume $F$ from DINO has been verified it has the potential in unsupervised semantic segmentations (Hamilton

et al., 2022). We next explore how to learn a segmentation field $s$ by leveraging $F$. Inspired by CRF and STEGO (Hamilton et al., 2022) where they refine the initial predictions using color or feature-correlated regions in the 2D image. We propose to append an extra segmentation branch to predict the segmentation field, formulating segmentation correspondence volume by leveraging its predicted segmentation logits using the same rule with Equation 2. Then, we construct the appearance-segmentation correlation aims to enforce the elements of $s$ and $s'$ closer if $f$ and $f'$ are tightly coupled, where the expression with and without the superscript indicates two different views. The volume correlation can be achieved via an element-wise multiplication between $S$ and $F$, and thereby, we have the appearance contrastive loss $\mathcal{L}_{app}$:

$$\mathcal{C}_{app}(\boldsymbol{r}, b) = - \sum_{hwh'w'} (F_{hwh'w'} - b) S_{hwh'w'} \tag{3}$$

$$\mathcal{L}_{app} = \lambda_{id} \mathcal{C}_{app}(\boldsymbol{r}_{id}, b_{id}) + \lambda_{neg} \mathcal{C}_{app}(\boldsymbol{r}_{neg}, b_{neg}) \tag{4}$$

where $S_{hwh'w'} = \sum_c \frac{s_{chw}}{|s_{hw}|} \frac{s'_{ch'w'}}{|s'_{h'w'}|}$ indicates the segmentation correspondence volume between two views, $\boldsymbol{r}$ is the cast ray fed into NeRF, $b$ is a hyper-parameter to control the positive and negative pressure. $\lambda_{id}$ and $\lambda_{neg}$ indicate loss force between identity pairs (positive) and distinct pairs (negative). The intuition behind the above equation is that minimizing $\mathcal{L}_{app}$ with respect to $S$, to enforce entries in segmentation field $s$ to be large when $F - b$ are positive items and pushes entries to be small if $F - b$ are negative items.

**Discover Patch Relationships** To construct Equation 4, we build a cosine similarity matrix to effectively discover the positive/negative pairs of given patches. For each matrix, we take $N$ randomly selected patches as inputs and adopt a pre-trained DINO-ViT to extract meaningful representations. We use the [CLS] token from ViT architecture to represent the semantic features of each patch and obtain $N$ positive pairs by the diagonal entries and $N$ negative pairs by querying the lowest score in each row. An example using three patches from different views is shown in Figure 3. Similar to Tumanyan et al. (2022), we observe that the [CLS] token from a self-supervised pre-trained ViT backbone can capture high-level semantic appearances and can effectively discover similarities between patches during the proposed end-to-end optimization process.

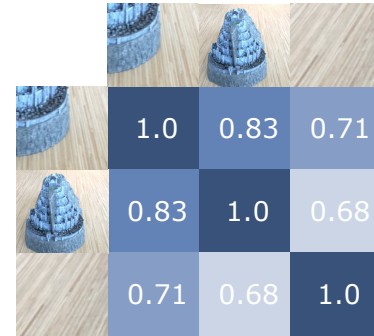

Figure 3: Cosine similarity matrix calculated on scene *Fortress*.

### 3.3 CROSS VIEW GEOMETRY CORRESPONDENCE

Constructing the "appearance-segmentation correlation" enables the clustered features with similar appearance together. However, appearance cue alone may cause spatial discontinuities, as DINO-ViT may overfocus to capture semantic parts rather than making the clusters spatial smooth (see Figure 7). Therefore, we propose geometric correlation volume to penalize discontinuities between neighboring points, by formulating the attractive/repulsive force using point-wise distance.

**Geometry Correspondence across Views** Apart from distilling the visual feature from DINO into the segmentation field $s$, we propose to leverage the density field that already exists in NeRF models to formulate a new geometry contrastive loss to encourage spatial coherence. Specifically, given a batch of $M$ cast ray $\boldsymbol{r}$ as NeRF's input, we can obtain the density field of size $M \times K$ where $K$ indicates the number of sampled points along each ray. By accumulating the discrete bins along each ray, we can roughly represent the density field as a single 3D point:

$$\boldsymbol{p} = \boldsymbol{r}_o + \boldsymbol{r}_d \cdot D \tag{5}$$

$$D(\boldsymbol{r}) = \sum_{k=1}^{K} T(k)(1 - \exp(-\sigma_k \delta_k)) t_k \tag{6}$$

where $\boldsymbol{p}$ is the accumulated 3D point along the ray, $D$ is the estimated depth value of the corresponding pixel index. Inspired by Point Transformer (Zhao et al., 2021) which uses point-wise distance as representation, we utilize the estimated point position as a geometry cue to formulate a new geometry

Table 1: Quantitative comparison of the novel view synthesis and object segmentation of LLFF dataset on the scenes *Flower* and *Fortress*.

| Scene "*Flower*" | PSNR ↑ | SSIM ↑ | LPIPS ↓ | NV-ARI ↑ | IoU(BG) ↑ | IoU(FG) ↑ | mIoU ↑ |
|---|---|---|---|---|---|---|---|
| IEM (Savarese et al., 2021) | - | - | - | 0.2666 | 0.7123 | 0.4267 | 0.5695 |
| DOCS (Li et al., 2018) | - | - | - | 0.0097 | 0.4824 | 0.2461 | 0.3643 |
| DINO+CoSeg (Amir et al., 2021) | - | - | - | 0.5946 | 0.9036 | 0.5961 | 0.7498 |
| NeRF-SOS (Ours) | 25.96 | 0.7717 | 0.1502 | 0.9529 | 0.9869 | 0.9503 | 0.9686 |
| Semantic-NeRF (Zhi et al., 2021) (Supervised) | 25.52 | 0.7500 | 0.1739 | 0.9104 | 0.9743 | 0.9090 | 0.9417 |
| Scene "*Fortress*" | PSNR ↑ | SSIM ↑ | LPIPS ↓ | NV-ARI ↑ | IoU(BG) ↑ | IoU(FG) ↑ | mIoU ↑ |
| IEM (Savarese et al., 2021) | - | - | - | 0.3700 | 0.7799 | 0.4526 | 0.6163 |
| DOCS (Li et al., 2018) | - | - | - | 0.7412 | 0.9329 | 0.7265 | 0.8297 |
| DINO+CoSeg (Amir et al., 2021) | - | - | - | 0.9503 | 0.9886 | 0.9395 | 0.9640 |
| NeRF-SOS (Ours) | 29.78 | 0.8517 | 0.1079 | 0.9802 | 0.9955 | 0.9751 | 0.9853 |
| Semantic-NeRF (Zhi et al., 2021) (Supervised) | 29.78 | 0.8578 | 0.0906 | 0.9838 | 0.9963 | 0.9799 | 0.9881 |

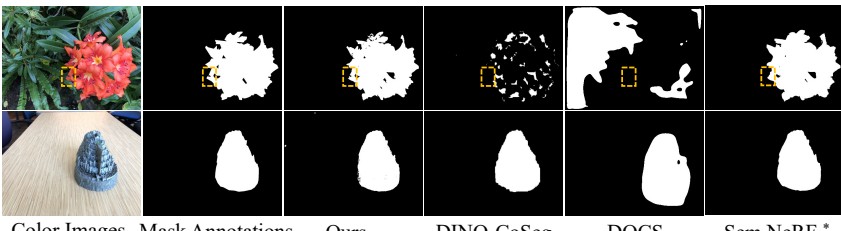

Color Images    Mask Annotations    Ours    DINO-CoSeg    DOCS    Sem.NeRF *

Figure 4: Qualitative results on scene *Flower* and *Fortress* of LLFF dataset. In the fourth column, DINO-CoSeg mistakenly matches several discrete patches, as DINO has higher activation on just a few tokens, which may lead to view-inconsistent and disconnected co-segmentation results. ∗ superscript denotes the supervised method. DOCS and DINO-CoSeg are not able to perform novel view synthesis, and thus we perform rendering before segmentation using a vanilla NeRF.

level correspondence volume across views by measuring point-wise absolute distance:

$$G_{hwh'w'} = \sum_c \frac{1}{|g_{chw} - g'_{ch'w'}| + \epsilon} \tag{7}$$

where $g$ and $g'$ are the estimated 3D point positions in two random patches of different views, $c$ is 3, $(h, w)$ and $(h', w')$ denote the spatial location on feature tensor for $g$ and $g'$, respectively.

**Injecting Geometry Coherence into Segmentation Field** To inject the geometry cue from the density field to the segmentation field, we formulate segmentation correspondence volume $S$ and geometric correspondence volume $G$ using the same rule of Equation 3. By pulling/pushing positive/negative pairs for the geometry-segmentation correlation of Equation 8, we come up with a new geometry-aware contrastive loss $\mathcal{L}_{geo}$:

$$\mathcal{C}_{geo}(\boldsymbol{r}, b) = -\sum_{hwh'w'} (G_{hwh'w'} - b) S_{hwh'w'} \tag{8}$$

$$\mathcal{L}_{geo} = \lambda_{id}\mathcal{C}_{geo}(\boldsymbol{r}_{id}, b_{id}) + \lambda_{neg}\mathcal{C}_{geo}(\boldsymbol{r}_{neg}, b_{neg}) \tag{9}$$

Same as appearance contrastive loss, we find positive pairs and negative pairs via the pair-wise cosine similarity of the [CLS] tokens.

### 3.4 OPTIMIZING WITH STRIDE RAY SAMPLING

We adopt patch-wise ray casting during the training process, while we also leverage a *Stride Ray Sampling* strategy, similar to prior works (Schwarz et al., 2020b; Meng et al., 2021) to handle GPU memory bottleneck. Overall, we optimize the pipeline using a balanced loss function:

$$\mathcal{L} = \lambda_0\mathcal{L}_{photometric} + \lambda_1\mathcal{L}_{app} + \lambda_2\mathcal{L}_{geo}, \tag{10}$$

where $\lambda_0$, $\lambda_1$, and $\lambda_2$ are balancing weights.

## 4 EXPERIMENTS

### 4.1 EXPERIMENT SETUP

**Datasets** We evaluate all methods on four representative datasets: Local Light Field Fusion (LLFF) dataset (Mildenhall et al., 2019), BlendedMVS (Yao et al., 2020), CO3Dv2 (Reizenstein et al., 2021),

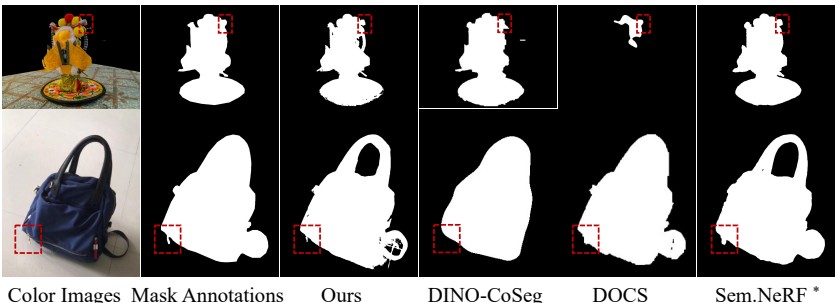

Color Images    Mask Annotations    Ours    DINO-CoSeg    DOCS    Sem.NeRF *

Figure 5: Novel view object segmentation results on object-centric datasets: BlendedMVS (the 1st row) and CO3Dv2 (the 2nd row). NeRF-SOS (the 3rd column) produces masks with finer details.

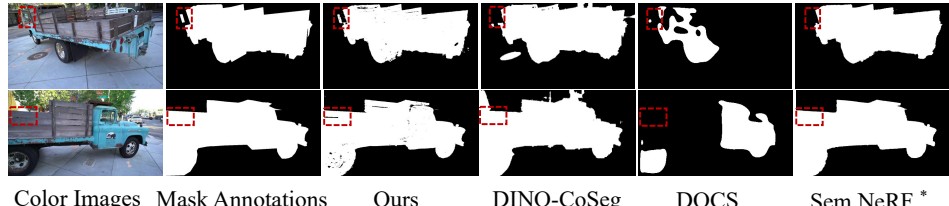

Color Images    Mask Annotations    Ours    DINO-CoSeg    DOCS    Sem.NeRF *

Figure 6: Novel view object segmentation results on unbounded scene *Truck*. NeRF-SOS (the 3rd column) produces more view-consistent masks than other self-supervised methods. It even generates finer details than supervised Semantic-NeRF++ (see the gaps between wooden slats in the top row and the side view mirror in the bottom row).

and Tank and Temples (T&T) dataset (Riegler & Koltun, 2020). Particularly, we use the forward-facing scenes {*Flower*, *Fortress*} from LLFF dataset, two object-centric scenes from BlendedMVS dataset, two common objects {*Backpack*, *Apple*} captured by video sequences from CO3Dv2 dataset, and unbounded scene *Truck* from hand-held 360° capture large-scale Tank and Temples dataset. We choose these representative scenes because they contain at least one common object among most views. We manually labeled all views as a binary mask to provide a fair comparison for all methods and used them to train Semantic-NeRF. Foreground objects appearing in most views are labeled as 1, while others are labeled as 0. We train and test all methods with the original image resolutions.

**Training Details**  We first implement the collaborative contrastive loss upon the original NeRF (Mildenhall et al., 2020a). In training, we first train NeRF-SOS without segmentation branch following the NeRF training recipe (Mildenhall et al., 2020b) for 150k iterations. Next, we load the weight and start to train the segmentation branch alone using the stride ray sampling for another 50k iterations. Model weights except the segmentation branch are kept frozen in the second phase. The loss weights $\lambda_0$, $\lambda_1$, $\lambda_2$, $\lambda_{id}$, and $\lambda_{neg}$ are set 0, 1, 0.01, 1 and 1 in training the segmentation branch. The segmentation branch is formulated as a four-layer MLP with ReLU as the activation function. The dimensions of hidden layers and the number of output layers are set as 256 and 2, respectively. The segmentation results are based on K-means clustering on the segmentation logits. We train Semantic-NeRF (Zhi et al., 2021) for 200k in total for fair comparisons. We randomly sample eight patches from different viewpoints (a.k.a batch size $N$ is 8) in training. The patch size of each sample is set as $64 \times 64$, with the patch stride as 6. We use the official pre-trained DINO-ViT in a self-supervised manner on ImageNet dataset as our 2D feature extractor. The pre-trained DINO backbone is kept frozen for all layers during training. All hyperparameters are carefully tuned by a grid search, and the best configuration is applied to all experiments. All models are trained on an NVIDIA RTX A6000 GPU with 48 GB memory. We reconstruct $N$ positives and $N$ negatives pairs on the fly during training, given $N$ rendered patches. More details can be found in the appendix.

**Metrics**  We adopt the Adjusted Rand Index in novel views as a metric to evaluate the clustering quality, noted as NV-ARI. We also adopt mean Intersection-over-Union to measure segmentation quality for both object and background, as we set the clusters with larger activation as foreground by DINO. To evaluate the rendering quality, we follow NeRF (Mildenhall et al., 2020a), adopting peak signal-to-noise ratio (PSNR), the structural similarity index measure (SSIM) (Wang et al., 2004), and learned perceptual image patch similarity (LPIPS) (Zhang et al., 2018) as evaluation metrics.

Table 2: Quantitative evaluation of the novel view synthesis and object segmentation on BlendedMVS and CO3Dv2 datasets, with several 2D object discovery frameworks and the supervised Semantic-NeRF. Results on each dataset are averaged on all scenes.

| BlendedMVS | PSNR ↑ | SSIM ↑ | LPIPS ↓ | NV-ARI ↑ | IoU(BG) ↑ | IoU(FG) ↑ | mIoU ↑ |
|---|---|---|---|---|---|---|---|
| IEM (Savarese et al., 2021) | - | - | - | 0.1339 | 0.5615 | 0.3715 | 0.4665 |
| DOCS (Li et al., 2018) | - | - | - | 0.7031 | 0.9183 | 0.7030 | 0.8107 |
| DINO+CoSeg (Amir et al., 2021) | - | - | - | 0.9074 | 0.9692 | 0.917 | 0.9431 |
| NeRF-SOS (Ours) | 23.86 | 0.8089 | 0.1288 | 0.9280 | 0.9756 | 0.9347 | 0.9552 |
| Semantic-NeRF (Zhi et al., 2021) (Supervised) | 23.84 | 0.8080 | 0.1339 | 0.9359 | 0.9803 | 0.9391 | 0.9597 |
| CO3Dv2 | PSNR ↑ | SSIM ↑ | LPIPS ↓ | NV-ARI ↑ | IoU(BG) ↑ | IoU(FG) ↑ | mIoU ↑ |
| IEM (Savarese et al., 2021) | - | - | - | 0.4784 | 0.7983 | 0.5708 | 0.6845 |
| DOCS (Li et al., 2018) | - | - | - | 0.8918 | 0.9684 | 0.8928 | 0.9307 |
| DINO+CoSeg (Amir et al., 2021) | - | - | - | 0.8199 | 0.9559 | 0.8222 | 0.8891 |
| NeRF-SOS (Ours) | 30.37 | 0.9358 | 0.073 | 0.9381 | 0.9813 | 0.9401 | 0.9607 |
| Semantic-NeRF (Zhi et al., 2021) (Supervised) | 31.17 | 0.9405 | 0.0603 | 0.9399 | 0.9821 | 0.9410 | 0.9615 |

Table 3: Quantitative results of the object segmentation results on outdoor unbounded scene *Truck*, with several 2D object discovery frameworks and the supervised Semantic-NeRF.

| Scene "*Truck*" | PSNR ↑ | SSIM ↑ | LPIPS ↓ | NV-ARI ↑ | IoU(BG) ↑ | IoU(FG) ↑ | mIoU ↑ |
|---|---|---|---|---|---|---|---|
| IEM (Savarese et al., 2021) | - | - | - | 0.3341 | 0.6791 | 0.5998 | 0.6395 |
| DOCS (Li et al., 2018) | - | - | - | 0.1517 | 0.6845 | 0.2463 | 0.4654 |
| DINO+CoSeg (Amir et al., 2021) | - | - | - | 0.8571 | 0.9408 | 0.9080 | 0.9244 |
| NeRF-SOS (Ours) | 22.20 | 0.7000 | 0.2691 | 0.9207 | 0.9689 | 0.9455 | 0.9572 |
| Semantic-NeRF++ (Zhi et al., 2021) (Supervised) | 21.08 | 0.6350 | 0.4114 | 0.9674 | 0.9869 | 0.9782 | 0.9826 |

## 4.2 COMPARISONS

**Self-supervised Object Segmentation on LLFF** We build NeRF-SOS on the vanilla NeRF (Mildenhall et al., 2020a) to validate its effectiveness on LLFF datasets. Two groups of current object segmentation are adopted for comparisons: $i$. NeRF-based methods, including our NeRF-SOS, and supervised Semantic-NeRF (Zhi et al., 2021) trained with annotated masks; $ii$. image-based object co-segmentation methods: DINO-CoSeg (Amir et al., 2021) and DOCS (Li et al., 2018); and $iii$. single-image based unsupervised segmentation: IEM (Savarese et al., 2021) follows CIS (Yang et al., 2019) to minimize the mutual information of foreground and background. As image-based segmentation methods cannot generate novel views, we pre-render the new views using NeRF and construct image pairs between the first image in the test set with others for DINO-CoSeg (Amir et al., 2021) and DOCS (Li et al., 2018). Evaluations on IEM also use pre-rendered color images.

Quantitative comparisons against other segmentation methods are provided in Table 1, together with qualitative visualizations shown in Figure 4. These results convey several observations to us: **1).** NeRF-SOS consistently outperforms image-based co-segmentation in evaluation metrics and view consistency. **2).** Compared with SoTA supervised NeRF segmentation method (Semantic-NeRF (Zhi et al., 2021)), our method effectively segments the object within the scene and performs on par in both evaluation metrics and visualization.

**Self-supervised Object Segmentation on Object-centric Scenes** For the object-centric datasets BlendedMVS and CO3Dv2, we uniformly select 12.5% of total images for testing. CO3Dv2 provides coarse segmentation maps using PointRend (Kirillov et al., 2020) while parts of the annotations are missing. Therefore, we manually create faithful binary masks for training the Semantic-NeRF and evaluations. As we can see in Table 2 and Figure 5, our self-supervised NeRF method consistently surpasses other 2D methods. We deliver more details comparisons in our supplementary materials.

**Self-supervised Object Segmentation on Unbounded Scene** To test the generalization ability of the proposed collaborative contrastive loss, we implement it on NeRF++ (Zhang et al., 2020) to test with a more challenging unbounded scene. Here, we mainly evaluate all previously mentioned methods on scene *Truck* as it is the only scene captured surrounding an object provided by NeRF++. We re-implement Semantic-NeRF using NeRF++ as the backbone model for unbounded setting, termed Semantic-NeRF++. Compared with supervised Semantic-NeRF++, NeRF-SOS achieves slightly worse results on quantitative metrics (see Table 3). Yet from the visualizations, we see that NeRF-SOS yields quite decent segmentation quality. For example, **1).** In the first row of Figure 6, NeRF-SOS can recognize the side view mirror adjacent to the truck. **2).** In the second row of Figure 6, NeRF-SOS can distinguish the apertures between the wooden slats as those apertures have distinct depths than the neighboring slats, thanks to the geometry-aware contrastive loss. Further, we show the 3-center clustering results on the distilled segmentation field in Figure 8.

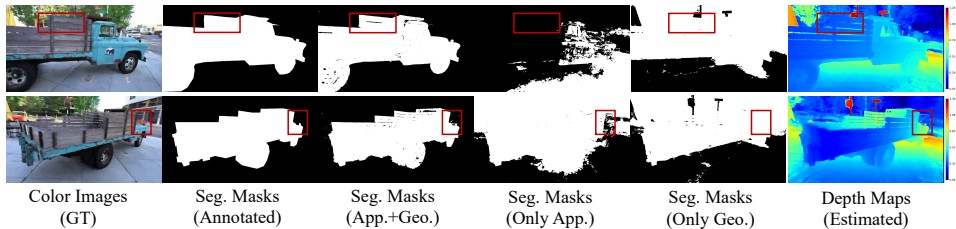

| Color Images (GT) | Seg. Masks (Annotated) | Seg. Masks (App.+Geo.) | Seg. Masks (Only App.) | Seg. Masks (Only Geo.) | Depth Maps (Estimated) |

Figure 7: Object segmentations using three loss variants are shown in columns 3, 4, and 5: the collaborative loss (APP.+Geo.), appearance-only loss (App.); geometric-only loss (Geo.).

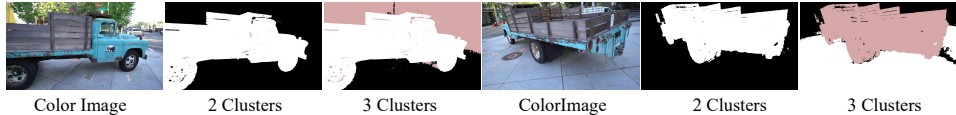

| Color Image | 2 Clusters | 3 Clusters | ColorImage | 2 Clusters | 3 Clusters |

Figure 8: Qualitative results on scene *Truck* with different cluster centers on its distilled segmentation field. Note that the cross-view visualized colors of multiple-center clustering are not corresponding to the subject ID, as we perform unsupervised clustering.

Table 4: Experiments on multiple NeRF-SOS variants. We show the results of joint training of the NeRF and contrastive loss in the first row, NeRF-SOS with ResNet50 as feature extractor in the second row, and our final model in the last row.

| Scene *"Flower"* | PSNR ↑ | SSIM ↑ | LPIPS ↓ | NV-ARI ↑ | IoU(BG) ↑ | IoU(FG) ↑ | mIoU ↑ |
|---|---|---|---|---|---|---|---|
| NeRF-SOS (Joint training) | 16.96 | 0.4585 | 0.7238 | 0.1961 | 0.7951 | 0.2220 | 0.5091 |
| NeRF-SOS (ResNet) | 25.96 | 0.7717 | 0.1502 | 0.8672 | 0.9421 | 0.8827 | 0.9124 |
| NeRF-SOS (Two-stage training) | 25.96 | 0.7717 | 0.1502 | 0.9529 | 0.9869 | 0.9503 | 0.9686 |

### 4.3 ABLATION STUDY

**Impact of the Collaborative Contrastive Loss**   To study the effectiveness of the collaborative contrastive loss, we adopt two baseline models by only using appearance contrastive loss or geometric contrastive loss on NeRF++ backbone. As shown in Figure 7, we observe that the segmentation branch failed to cluster spatially continuous objects without geometric constraints (mIoU: 0.5029). Similarly, without visual cues, the model lost the perception of the central object (mIoU: 0.5516). Our full model constructs precise clusters with spatial coherence (mIoU: 0.9689).

**Joint Training with NeRF Optimization**   To demonstrate the advantages of two-stage training, we conduct an ablation study by jointly optimizing vanilla NeRF rendering loss and the proposed two-level collaborative contrastive loss. As shown in Table 4, both the novel view synthesis quality and the segmentation quality significantly decreased when we optimize the two losses together. We conjecture the potential reason to be the fact that the optimization process of NeRF training is affected by the conflicting update directions, the reconstruction loss and the contrastive loss, which remains a notorious challenge in the multi-task learning area (Yu et al., 2020).

**CNN-based Backbone for Feature Extraction**   DINO-ViT firstly concludes that ViT architecture can extract stronger semantic information than ConvNets when being self-supervised trained. To study its effect on discovering the semantic layout of scenes, we apply self-supervised ResNet50 (He et al., 2020) as backbones. The results in the second row of Table 4 imply that the ViT architecture is more suitable for our NeRF object segmentation in both expressiveness and pair-selection perspectives.

## 5 CONCLUSION, DISCUSSION OF LIMITATION

In this paper, we introduce NeRF-SOS, a self-supervised framework that learns object segmentation for any view in complex real-world scenes. NeRF-SOS proposes a collaborative contrastive loss in both the appearance and geometry levels. Comprehensive experiments are conducted on four different types of datasets with state-of-the-art image-based object (co-)segmentation frameworks and fully supervised Semantic-NeRF. The results show that NeRF-SOS consistently outperforms image-based methods and sometimes generates finer segmentation details than its supervised counterparts. However, similar to other scene-specific NeRF methods, one limitation of NeRF-SOS is that it cannot segment across scenes, which we plan to explore in future work.

## ACKNOWLEDGEMENT

We would like to express our gratitude to Xinhang Liu from HKUST and Zhongzheng Ren from UIUC for their invaluable contribution to the experimental work presented in this paper. Their expertise and time proved to be immensely helpful in conducting the necessary comparisons with their methods, especially since their codes were not publicly available. We greatly appreciate their generosity and support throughout the research process.

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

## A APPENDIX

### A.1 ADDITIONAL TRAINING DETAILS

**Implementation of the Patch Selection**  We reconstruct positive and negative pairs on the fly during training. Given $N$ rendered patches from $N$ different viewpoints in training, we fed the patches into the DINO-ViT and obtained the [CLS] tokens. Next, we compute a $N \times N$ similarity matrix using the cosine similarity with the [CLS] tokens. The negative pairs are selected from the pair with the lowest similarity in each row; the positive pairs are set as the identity pairs. Overall, $2N$ pairs ($N$ positives + $N$ negatives) are formulated per iteration to compute the collaborative contrastive loss.

**Implementation of Stride Ray Sampling**  Neural radiance field casts a number of rays (typically not adjacent) from the camera origin, intersecting the pixel, to generate input 3D points in the viewing frustum. Our model requires patch-wise rendering of size $(P, P)$ to formulate the collaborative contrastive loss. However, we can only render a patch less than $64 \times 64$ in each view due to GPU memory bottleneck (Garbin et al., 2021). Thus, it hardly covers a sufficient receptive field to capture the global context, using the pre-trained DINO. To solve this problem, we adopt a *Strided Ray Sampling* strategy (Schwarz et al., 2020b; Meng et al., 2021), to enlarge the receptive field of the patches while keeping computational cost fixed. Specifically, instead of sampling a patch of adjacent locations $P \times P$, we sample rays with an interval $k$, resulting in a receptive field of $(P \times k) \times (P \times k)$.

**Hyperparameters Selection**  The hyperparameters of NeRF-SOS on different datasets are shown in Table 5. We adopt the number of $b^{knn}$ and $b^{self}$ in (Hamilton et al., 2022) as the hyperparameter for our appearance level loss ($b^{neg}$ and $b^{id}$, respectively.). We share the appearance level hyperparameters across all datasets. We set the weights of $b^{neg}$ and $b^{id}$ in the geometry level loss with physical intuition. For example, since the radius of the foreground object in LLFF datasets is roughly 0.5 meters, we set the $b^{neg}$ and $b^{id}$ to be 0.5 and 3, respectively. Analogously, for the unbounded scene (e.g., scene *Truck*), the $b^{neg}$ and $b^{id}$ are set to be 1 and 5, respectively.

Table 5: Hyperparameters of NeRF-SOS on different datasets. We share the hyperparameters of appearance level for all datasets while we set the $b^{neg}$ and $b^{id}$ for geometry level loss with physical intuition.

| parameter name | LLFF | BlendedMVS | CO3Dv2 | Tank and Temples |
|---|---|---|---|---|
| $b_{id}(\mathcal{L}_{geo})$ | 0.50 | 0.12 | 0.25 | 1.00 |
| $b_{neg}(\mathcal{L}_{geo})$ | 3.00 | 0.60 | 1.00 | 5.00 |

**Self-supervised Learned 2D Representations**  We adopt DINO-ViT (Caron et al., 2021) as our feature extractor for distillation. The training process of DINO-ViT largely simplifies self-supervised learning by applying a knowledge distillation paradigm (Hinton et al., 2015) with a momentum encoder (He et al., 2020), where the model is simply updated by a cross-entropy loss.

### A.2 HUMAN ANNOTATION DETAILS

We use the publicly available annotation tool: (labelme) for the foreground and background annotation. To be specific, we annotate all training and testing views of different scenes using multiple-polygon, extract the polygons and convert them to binary masks. The masks of scene *Flower* are included in our supplementary and we provide usage guidelines in README. All annotated data will be made public.

### A.3 ADDITIONAL EXPERIMENTS

**Comparisons with Semantic-NeRF using Sparse Label**  As Semantic-NeRF ables to perform label propagation with sparse annotation, we simulate sparse user annotation by randomly applying {1, 1%, 5%, 10%} foreground annotated object pixels while leaving the rest unlabeled. We can see from Figure 9, the foreground boundaries are gradually refined when more annotations are included, which

Table 6: Comparisons with Semantic-NeRF using sparse labels. Results are calculated on scene *Fortress*

| Scene "*Fortress*" | User Click ↓ | NV-ARI ↑ | IoU(BG) ↑ | IoU(FG) ↑ | mIoU ↑ |
|---|---|---|---|---|---|
| Semantic NeRF | 1 | 0.9615 | 0.9812 | 0.9528 | 0.967 |
| Semantic NeRF | 1% | 0.9731 | 0.9938 | 0.9667 | 0.9803 |
| Semantic NeRF | 5% | 0.9783 | 0.9950 | 0.9731 | 0.9841 |
| Semantic NeRF | 10% | 0.9788 | 0.9952 | 0.9735 | 0.9843 |
| Semantic NeRF | 100% | 0.9838 | 0.9963 | 0.9799 | 0.9881 |
| NeRF-SOS | 0 | 0.9802 | 0.9955 | 0.9751 | 0.9853 |

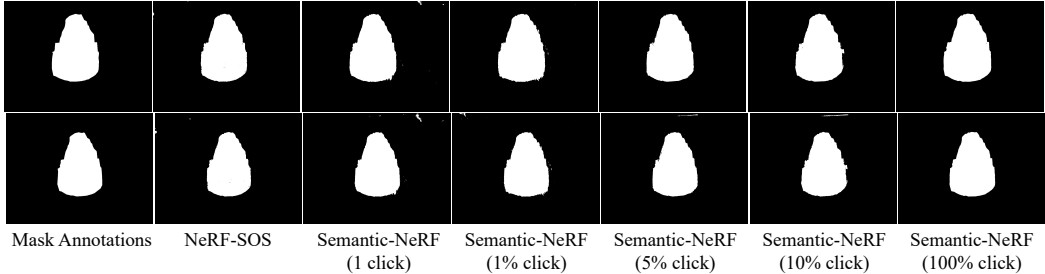

| Mask Annotations | NeRF-SOS | Semantic-NeRF (1 click) | Semantic-NeRF (1% click) | Semantic-NeRF (5% click) | Semantic-NeRF (10% click) | Semantic-NeRF (100% click) |

Figure 9: Visual comparisons among NeRF-SOS and Semantic-NeRF with sparse annotations.

Table 7: Comparisons among several NeRF-based methods for object segmentation.

| LLFF Dataset | Supervision Type | NV-ARI ↑ | IoU(BG) ↑ | IoU(FG) ↑ | mIoU ↑ |
|---|---|---|---|---|---|
| RFP | Self-supervised | 0.9267 | 0.9812 | 0.9178 | 0.9495 |
| NeRF-SOS | Self-supervised | 0.9665 | 0.9912 | 0.9627 | 0.9769 |
| NVOS | Weak-supervised | 0.9217 | 0.9793 | 0.9145 | 0.9469 |
| ObjectNeRF | Supervised | 0.9666 | 0.9909 | 0.9639 | 0.9774 |

is consistent with reported results in the original paper of Semantic-NeRF. However, we can see from Table 6, all sparse annotation experiments show insufficient accurate salient foreground segmentation, compared with dense annotated Semantic-NeRF. Whereas NeRF-SOS performs comparably with the dense annotation counterpart.

**Comparisons with other NeRF-based Object Segmentation**    Label-free NeRF segmentation on real-world scenes remains a challenging problem. ObjectNeRF (Yang et al., 2021) appends an object branch input with object feature and object code, with object-level supervision (a.k.a. 2D instance masks) to enable object manipulation after training. Although Neural Volumetric Object Selection (NVOS) (Ren et al., 2022) solves interactive weak-supervised object segmentation, it requires users to provide several scribbles for supervision. The concurrent work, RFP (Liu et al., 2022), tackles the label-free NeRF segmentation on real-world scenes, but there is still room for their segmentation accuracy. To handle the high-quality label-free NeRF segmentation on real-world scenes, NeRF-SOS leverages the proposed collaborative contrastive loss to do self-supervised object segmentation. Experiments in the following Tables 7 and Figure 10 demonstrate our label-free method consistently outperforms RFP and weakly-supervised NVOS.

**Qualitative Visualization on More Views**    Qualitative comparisons on LLFF dataset can be found in Figure 11 and Figure 12, respectively. Qualitative comparisons on BlendedMVS dataset can be found in Figure 13 and Figure 14, respectively. Qualitative comparisons on CO3Dv2 dataset can be found in Figure 15 and Figure 16, respectively. Qualitative comparisons on Tank and Temple dataset can be found in Figure 17. Here, we visualize three different views to show the segmentation consistency across views. Visualized video can be found in the supplementary.

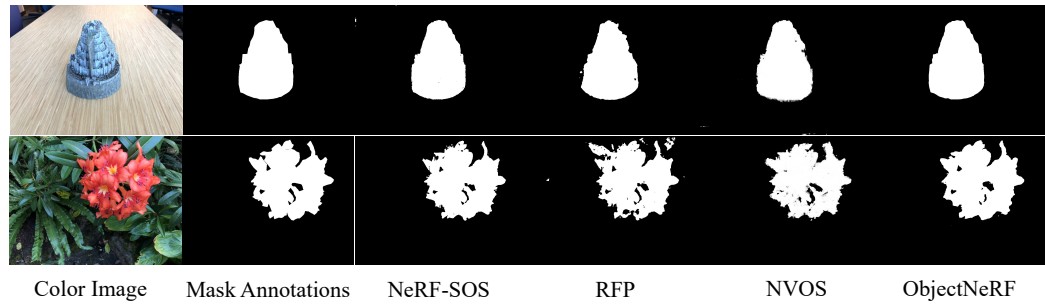

Color Image    Mask Annotations    NeRF-SOS    RFP    NVOS    ObjectNeRF

Figure 10: Visual comparisons of the segmentation masks among NeRF-SOS (self-supervised), RFP (self-supervised), NVOS (weakly-supervised), and ObjectNeRF (fully-supervised).

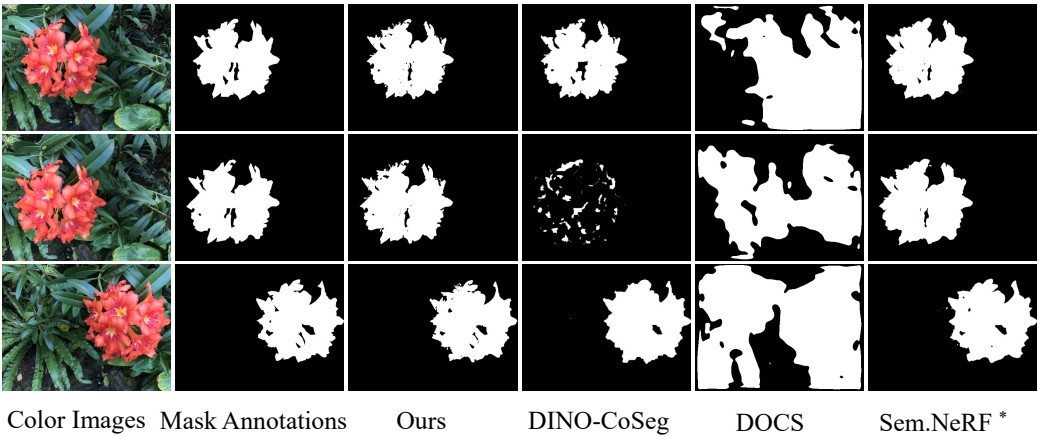

Color Images  Mask Annotations    Ours    DINO-CoSeg    DOCS    Sem.NeRF [*]

Figure 11: Novel view object segmentation results on scene *Flower* of LLFF dataset.

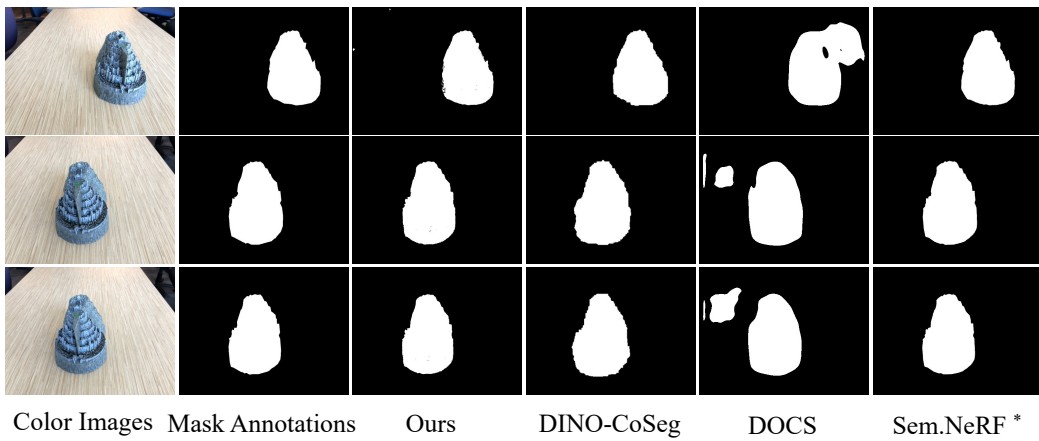

Color Images  Mask Annotations    Ours    DINO-CoSeg    DOCS    Sem.NeRF [*]

Figure 12: Novel view object segmentation results on scene *Foretress* of LLFF dataset.

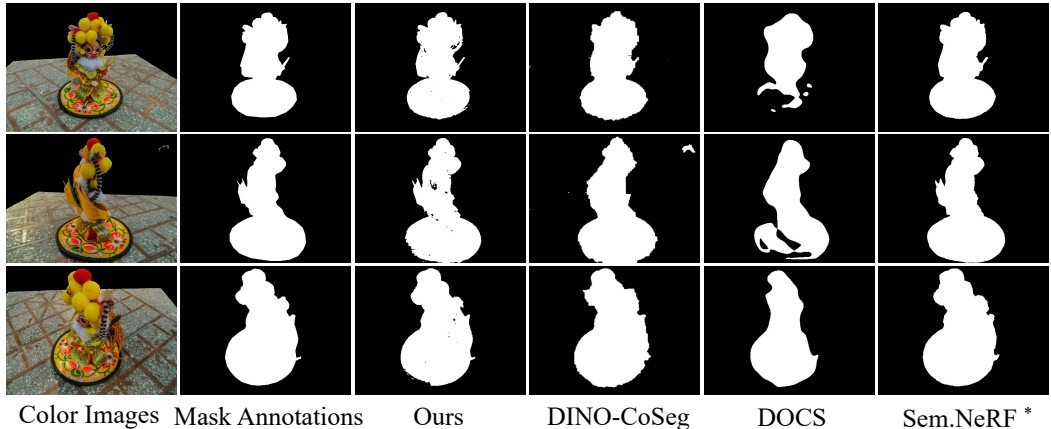

Color Images  Mask Annotations  Ours  DINO-CoSeg  DOCS  Sem.NeRF [*]

Figure 13: Novel view object segmentation results on scene *5a3* of BlendedMVS dataset.

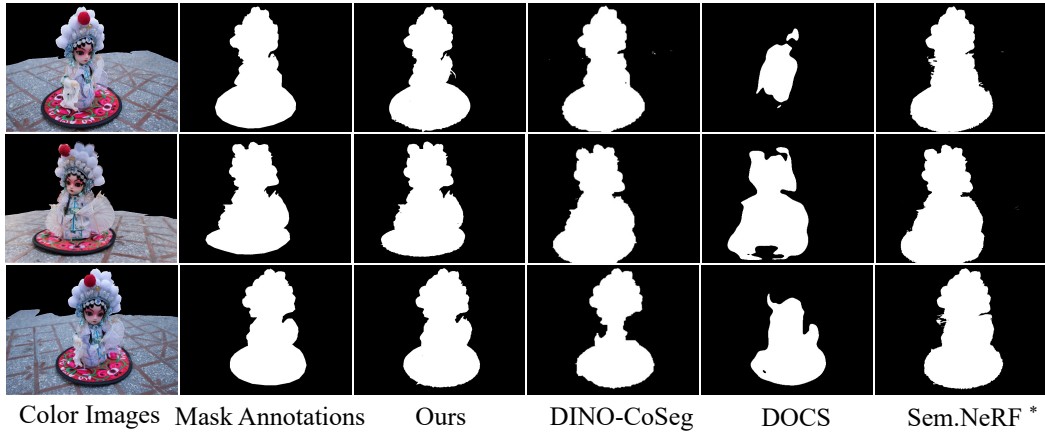

Color Images  Mask Annotations  Ours  DINO-CoSeg  DOCS  Sem.NeRF [*]

Figure 14: Novel view object segmentation results on scene *5a6* of BlendedMVS dataset.

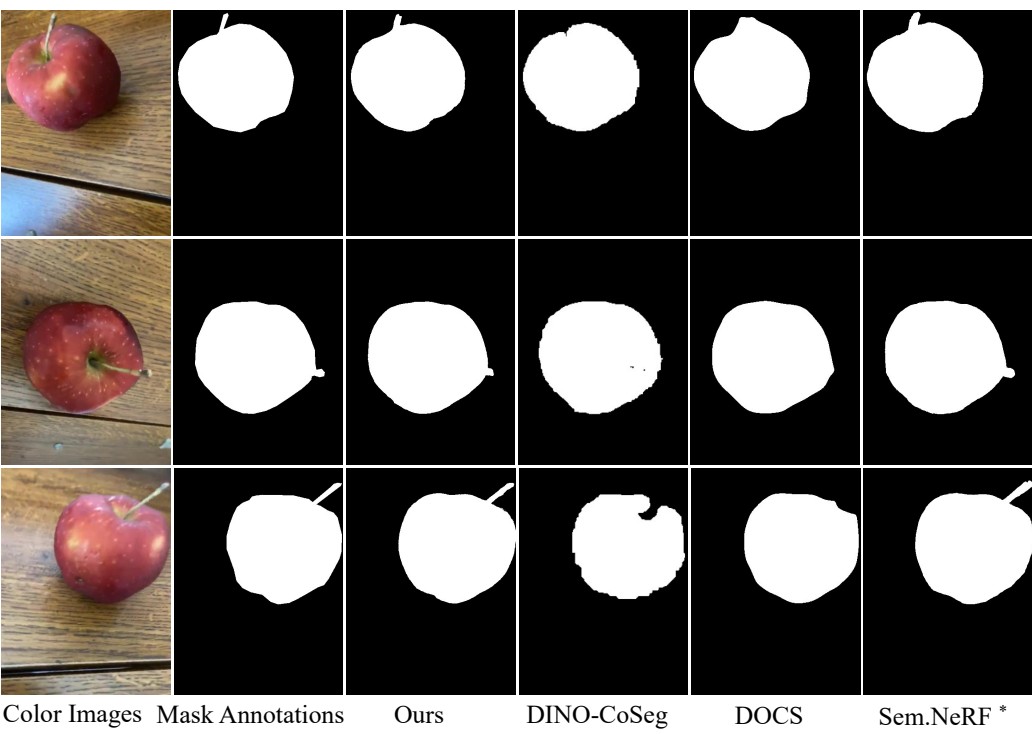

Color Images  Mask Annotations      Ours       DINO-CoSeg       DOCS       Sem.NeRF [*]

Figure 15: Novel view object segmentation results on scene *Apple* of CO3Dv2 dataset.

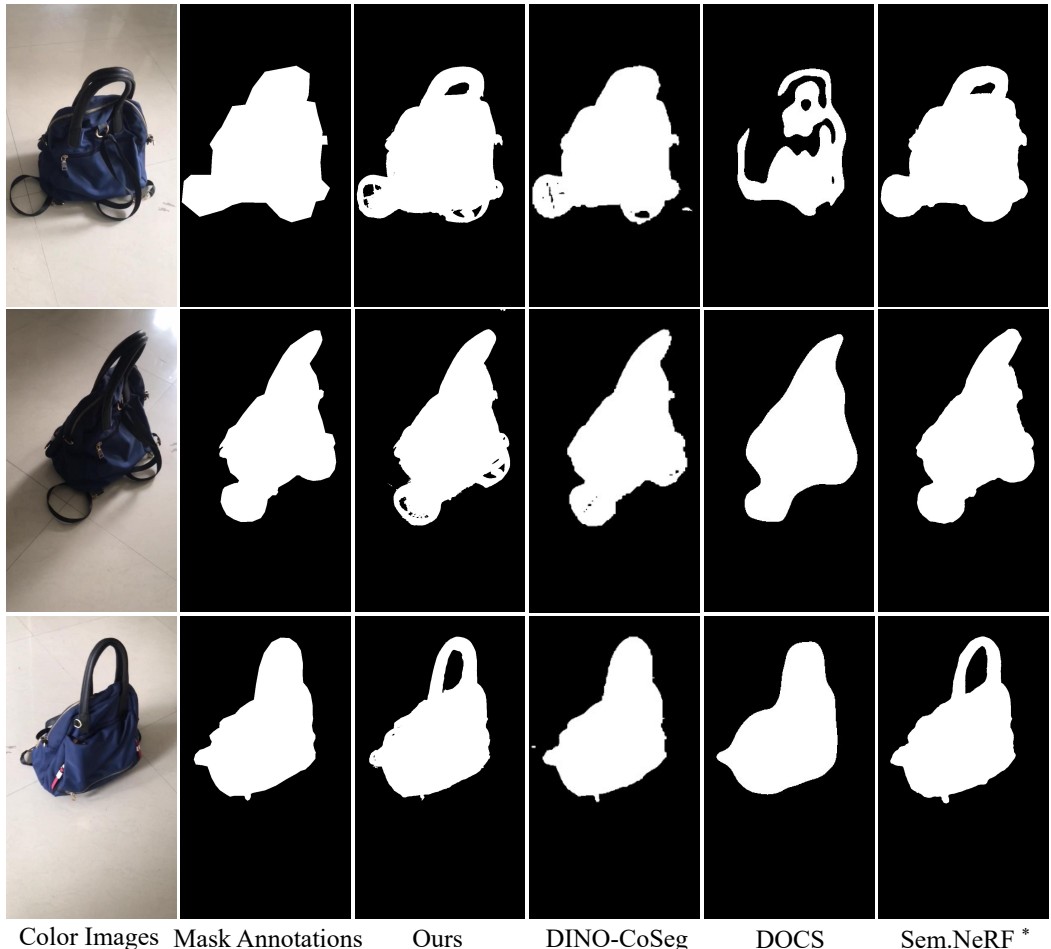

Figure 16: Novel view object segmentation results on scene *Backpack* of CO3Dv2 dataset.

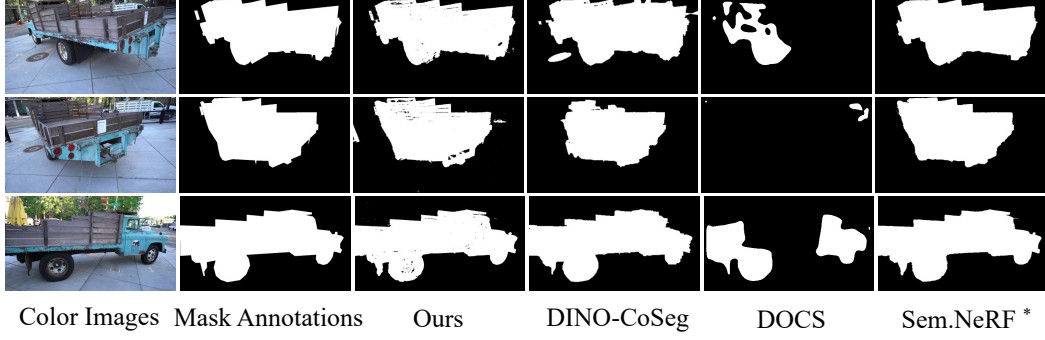

Figure 17: Novel view object segmentation results on scene *Truck* of Tank and Temples dataset.

