# OpenReview forum: "NeRF-SOS: Any-View Self-supervised Object Segmentation on Complex Scenes"
_ICLR.cc/2023/Conference — ICLR 2023 poster_

### Official Review · Reviewer_92aZ · 2022-10-19

**Confidence:** 4
**Correctness:** 3
**Technical Novelty And Significance:** 2
**Empirical Novelty And Significance:** 2
**Recommendation:** 6

**Clarity, Quality, Novelty And Reproducibility:**

I think this work is quite incremental as using a single network to regress density field, segmentation, and colors have already been proposed in previous papers. The appearance correlation volume is somewhat similar to part of the process in Neural Volumetric Object Selection which also use self-supervised learning to extract initial segmentation volume. The geometry constraint are also standard in multi-view stereo and has been demonstrated in Point transformer for point cloud segmentation. Overall, I feel this paper is just a combination of existing methods and then apply it to the NeRF segmentation.

**Strength And Weaknesses:**

Strength: It is a good attempt to extend previous cosegmentation method for NeRF setting. The proposed method is reasonable and the results are better than the compared methods.

Weaknesses: The proposed method is not the first work that deal with segmentation in NeRF. Some important references are missing. The experimental comparisons are incomplete as it miss the comparisons with some recent methods. Please check the following recent works:

Yu et al., Unsupervised Discovery of Object Radiance Fields, in ICLR, 2021 (missing comparisons)

Yang et al., Learning object-compositional neural radiance field for editable scene rendering, in ICCV 2021  (missing comparisons)

Ren et al., Neural Volumetric Object Selection, in CVPR 2022 (missing reference and comparison)

Smith et al., Unsupervised discovery and composition of object light fields, in ICLR, 2022. (missing reference and comparison)

Kundu et al., Panoptic Neural Fields: A Semantic Object-Aware Neural Scene Representation, in CVPR, 2022. (missing reference)

Liu et al., Unsupervised Multi-View Object Segmentation Using Radiance Field Propagation, in NeurIPS 2022. (This one should be considered as a concurrent work since NeurIPS conference is after ICLR deadline.)

**Summary Of The Paper:**

This paper presents a self supervised method for object segmentation in NeRF. The major idea is applying collaborative contrastive training in both appearance (radiance field) and geometry (density field) level in NeRF. It uses DINO (Amir et al., 2021)  as the basic backbone architecture to extract appearance correlation volume and extend it with geometry correspondence across views. The proposed method is evaluated on several NeRF dataset and show that it outperforms other cosegmentation methods, i.e. DINO-CoSeg and DOCS, that were designed for multiple image cosegmentation but not for NeRF setting.

**Summary Of The Review:**

Please check my above comments regarding missing references and comparisons, and the incremental novelty of the proposed method.

---

> ### Author Response · Authors · 2022-11-14
> **Response to Reviewer 92aZ**
>
> **[Q1]:** The novelty of the proposed method?
> **[A1]:** Label-free NeRF segmentation on real-world scenes remains a challenging problem. Although Neural Volumetric Object Selection (NVOS) [1] solves interactive weak-supervised object segmentation, it requires users to provide several scribbles for supervision. The concurrent work, RFP [2], tackles the label-free NeRF segmentation on real-world scenes, but there is still room for their segmentation accuracy. To handle the high-quality label-free NeRF segmentation on real-world scenes, we propose NeRF-SOS which is a fully self-supervised framework to extract the foreground object, by leveraging the proposed collaborative contrastive loss. Experiments in the following tables and Fig. 10 in our appendix demonstrate our label-free method consistently outperforms RFP and weakly-supervised NVOS. Results are averaged on LLFF datasets.
>
> | Methods/Metrics | Suervision Type | NV-ARI ↑ | IoU(BG) ↑  | IoU(FG) ↑ | mIoU ↑
> | :------------: | :----:| :----: | :----: | :----: | :----:
> |   RFP | Unsupervised| 0.9267 | 0.9812 |  0.9178 | 0.9495
> |    NeRF-SOS  | Unsupervised  | 0.9665 | 0.9912 | 0.9627 | 0.9769
> |    NVOS   | Weakly-supervised | 0.9217 | 0.9793 | 0.9145 | 0.9469
> |   Object NeRF    | Fully-Supervised  | 0.9666 | 0.9909 | 0.9639 | 0.9774
>
> **[Q2]:** The difference with Neural Volumetric Object Selection [1], especially in how to extract initial segmentation volume?
> **[A2]:** NVOS [1] lifts the users given 2D scribbles into 3D followed by an fg/bg classifier, combining multiple kinds of 3D features tensor including cost volume feature(standard in MVSNet variants), IBR voxel feature (representative in MPI), and positional voxel features (to indicate the 3D location) for its foreground/background classifier. However, the fg/bg training requires user-annotated scribbles for supervision, and the training process can not be regarded as self-supervised learning.
> Whereas NeRF-SOS aims to be free of users’ annotations, by smartly leveraging the semantic feature from self-supervised trained DINO, which can roughly reflect the salient foreground and background within a scene. Specifically, the “appearance correlation volume” in NeRF-SOS defines the semantic distance, to help the model minimize/maximize the intra/inter-cluster appearance by formulating attractive/repulsive force using positive/negative pairs.
> Although they are all named after “volume”, they are different in design philosophy and technique details.
>
> **[Q3]:** The difference of the geometry constraint with Multi-view Stereo and Point Transformer.
> **[A3]:** We are aware that geometric constraints are common in 3D vision community. However, they have different functions and purposes in different scenarios. For example, Point Transformer [4] uses point-wise distance as position encoding to describe the location of a token in a sequence. Learning-based MVSNet [3] formulates cost volume to constrain the homographic geometry. Whereas NeRF-SOS constructs “geometry level correspondence volume” to encourage spatial coherence of the clusters by penalizing discontinuities between neighboring points. The penalty is formulated by the attractive/repulsive force using positive/negative pairs. Although they have similar names, our geometric constraint is essentially different from MVS and Point Transformer.

---

> > ### Comment · Reviewer_92aZ · 2022-11-16
> > **Thanks for the additional comparisons**
> >
> > I am satisfied with the rebuttal from the authors and I will raise my score to 6: marginally above the acceptance threshold

---

> ### Author Response · Authors · 2022-11-14
> **Response to Reviewer 92aZ**
>
> **[Q4]:** The proposed method is not the first work that deals with segmentation in NeRF. Some important references are missing. The experimental comparisons are incomplete:
> **[A4]:** We have organized your mentioned papers into 4 groups, compared them with several works in the attached table, and discussed the difference with each group separately.
> * Unsupervised object discovery works on synthetic datasets [5][6].
>   * uORF[5] designs a set of latents (along with multiple conditional NeRF models) for the background and different objects for unsupervised 3D scene decomposition, and binds each slot to an object region via attention modules.
>   * COLF[6] addresses the high computational complexity of uORF. COLF proposes a light field compositor module to enable the composition of a set of object-centric light fields into the light field of the complete scene.
>   * Although uORF and COLF design prior-based frameworks for synthetic object decomposition. They require pre-training on large-scale datasets and perform experiments on specific categories. To extend NeRF segmentation to be totally label-free and on arbitrary categories,  we propose NeRF-SOS which does not require any change to the core NeRF model architecture. NeRF-SOS designs a collaborative contrastive loss on the parallel segmentation feature field to separate the foreground and background in real-world scenes for arbitrary categories.
>
> * Unsupervised real-world object segmentation using propagated radiance field [2].
>   * The concurrent work RFP [2] proposes a propagation strategy for individual objects’ radiance fields with a bidirectional photometric loss, to separate a scene into salient foreground and background. The object masks are refined with an iterative expectation-maximization algorithm.
>   * RFP enables label-free object segmentation in real-world scenes, for arbitrary categories. NeRF-SOS improve real-world segmentation accuracy by proposing a new collaborative contrastive loss to distill both appearance and geometric features in a self-supervised way. We can see from the table, NeRF-SOS boosts the segmentation accuracy from 0.9267 to 0.9665 on NV-ARI, and from 0.9495 to 0.9769 on mIoU. Note that, we have contacted the authors for comparisons as RFP is not open-source yet.
>
> * Object selection with annotated instance masks/scribbles [7][1].
>   * ObjectNeRF [7] appends an object branch input with object feature and object code, with object-level supervision (a.k.a. 2D instance masks) to enable object manipulation after training.
>   * Neural Volumetric Object Selection (NVOS) [1] formulates a new user-driven NeRF segmentation setting, where several scribbles are required from users to segment the object in the volumetric 3D scene representation with a post-processing step to remove floaters. To conduct 3D segmentation with sparse scribbles supervision, the authors incorporate 2D CNN feature, 3D cost volume feature aggregate by several 3D convolution layers, voxel feature from neural IBR volume, and positional voxel features. NVOS contains a classifier to indicate the foreground or background voxels using classification supervision from 2D scribbles.
>   * Real-world object segmentation using NeRF is a challenging task. ObjectNeRF leverages annotated object masks for supervised segmentation. NVOS solves interactive semi-supervised object segmentation, by lifting the given 2D scribbles into 3D along with multiple type volume features to classify the scribbles. Whereas NeRF-SOS is a fully self-supervised framework to extract the dominant object, by contrasting the segmentation feature using a collaborative loss.
> Label-free NeRF-SOS outperforms weakly-supervised NVOS (+4.48% on NV-ARI, +3.0% on mIoU), performs comparably with supervised ObjectNeRF (0.9769 vs. 0.9774 on mIoU).  Note that, we have contacted the authors of NVOS for comparisons as NVOS is not open-source yet.
>
> * Panoptic radiance field with annotations [8].
>   * PNF [8] proposes to represent each object instance using a separate MLP with meta-learning based initialization. The semantic and instance supervisions come from the off-the-shelf 2D semantic segmentor and 3D object detector. Therefore, the trained network can render color images and panoptic segmentations on new views.
>   * PNF aims to solve the supervised panoptic segmentation problem under NeRF formulation, on predefined categories. While NeRF-SOS enables segmenting arbitrary objects in general scenes using a self-supervised collaborative contrastive loss.
>
> We have added the missing reference and relevant discussions to our revised manuscript, following your suggestions.

---

> ### Author Response · Authors · 2022-11-14
> **Reference for the Response to Reviewer 92aZ**
>
> **Reference:**
> [1]. Ren et al., Neural Volumetric Object Selection
> [2]. Liu et al., Unsupervised Multi-View Object Segmentation Using Radiance Field Propagation
> [3]. Yao et al., MVSNet: Depth Inference for Unstructured Multi-view Stereo
> [4]. Zhao et al., Point Transformer
> [5]. Yu et al., Unsupervised Discovery of Object Radiance Fields
> [6]. Smith et al., Unsupervised discovery and composition of object light fields
> [7]. Yang et al., Learning object-compositional neural radiance field for editable scene rendering
> [8]. Kundu et al., Panoptic Neural Fields: A Semantic Object-Aware Neural Scene Representation

---

### Official Review · Reviewer_vb78 · 2022-10-25

**Confidence:** 4
**Correctness:** 3
**Technical Novelty And Significance:** 3
**Empirical Novelty And Significance:** 3
**Recommendation:** 8

**Clarity, Quality, Novelty And Reproducibility:**

The paper is a bit difficult to understand, especially the narrative around the correlation volumes. Improving the Grammar could help a bit (e.g., paragraph 2, last line - However, still remains a gap..., Page 4, last section: DINO has been verified it has the potential...). The technical details are listed without much explanation as to why they are necessary. To make this paper useful and reproducible, the authors should consider rewriting the technical section with more clarity.
Some details about the comparative methods are missing. Particularly, Semantic NeRF can be trained from sparse labels. What happens if only a single click is used to seed the object.

**Strength And Weaknesses:**

Since NeRF is trained under an i.i.d. assumption, there is little semantic correlation between the features across different views. Hence, the color or depth MLPs can't be directly used for semantic tasks. In this paper, a new semantic branch is added to the network that is trained by explicitly enforcing matches across views through collaborative contrastive losses. This is a novel idea, and it becomes feasible to extract the dominant object simply by clustering the logits of the semantic branch. This appears to be a really effective approach and is able to match the segmentation quality of semi-supervised methods such as the Semantic-NeRF.

Although the method claims to segment complex real-world scenes, the demonstrated examples are typically single object scenarios with no background noise. In most cases the objects are also centered and parallel to the visual plane. The authors mention that a density based correlation is required to distinguish between objects, but neither the qualitative nor the ablation results reflect the need for this additional loss. The results could be much more interesting if these design choices and other parameters such as the number of kmeans clusters were explained.

When a high-performance model such as DINO-ViT is used, it becomes difficult to tease apart the value of the core method vis-a-vis the strong model. The results of the proposed method with Resnet seem much less impressive that with DINO backbone. How much does the DINO model alone contribute to accurate segmentation?

The comparison to Semantic NeRF is also a bit misleading, since this model can segment only a single dominant object that's in the field of view, whereas Semantic NERF allows simultaneous segmentation of many class labels.



**Summary Of The Paper:**

The paper proposes a self-supervised object segmentation technique that operates on an optimized Neural Radiance Field. It couples NeRF reconstruction with DiNO-ViT feature extraction and models a collaborative contrastive loss function that helps parse the dominant object in the image. The results show that the segmentation accuracy is at par with the semi-supervised Semantic NeRF and outperforms other approaches.

**Summary Of The Review:**

The paper proposes a novel idea of object segmentation by coupling the NeRF color and depth features with DINO powered semantic segmentation features. It provides a nice demonstration that the dominant object in the NeRF model can be parsed in a full self-supervised manner. The details of how to go about it are a bit sketchy, and the lack of detailed ablation studies makes it harder to understand the merits and the inner workings of the correlation volumes. Adding more diverse results of complex real-world scenes, and teasing apart the results from the DINO model alone would make the results more interesting.

---

> ### Author Response · Authors · 2022-11-14
> **Response to Reviewer vb78**
>
> **[Q1]:** Explain the design choices of the “density based correlation” to distinguish between objects.
> **[A1]:** The introduction of “geometry-segmentation correlation” from the density field encourages spatial coherence of the distilled segmentation field. We can formulate an attractive or repulsive force via the positive or negative pairs to achieve this point. For example, we can observe the clusters with spatial discontinuities when we only use appearance correspondence (see the 1st row, 4th column of Fig. 7 in the main draft) . Such discontinuities are induced by the imprecise clusters, as the distilled features from DINO-ViT overfocus on capturing semantic object parts [1] while ignoring the cluster's spatial coherence.
> Driven by this problem, we design a “geometry-aware contrative loss” to make the distilled feature clusters spatial coherence, using the geometry-segmentation correlation. We have added the explanation in the revision.
>
> **[Q2]:** Parameters setting of K-means clustering?
> **[A2]:** We leverage the K-means algorithm integrated into the sklearn library [2] by setting n_clusters as 2, and init as k-means++. It will initialize two centroids and use k-means++ for the subsequent clustering operation.
>
> **[Q3]:** The dataset choices.
> **[A3]:** Label-free NeRF segmentation on real-world scenes remains a challenge. Several prior works [3,4] in unsupervised NeRF object decomposition can only segment synthetic data with limited categories, even with pre-training on large 3D datasets. Although Neural Volumetric Object Selection (NVOS) [5] solves interactive weak-supervised object segmentation, it requires users to provide several scribbles for supervision. Label-free NeRF segmentation on real-world complex scenes is not feasible until the concurrent work, RFP [6]. Similar to RFP, we also validate our method on several real-world NeRF datasets, including LLFF, CO3Dv2, BlendedMVS, and even on the unbounded Tank and Temples. We present the comparisons with RFP in the table below, and we can see NeRF-SOS performs better than RFP and NVOS on LLFF datasets, validating the effectiveness of our model design.
>
> | Methods/Metrics | Suervision Type | NV-ARI ↑ | IoU(BG) ↑  | IoU(FG) ↑ | mIoU ↑
> | :------------: | :----:| :----: | :----: | :----: | :----:
> |   RFP | Unsupervised| 0.9267 | 0.9812 |  0.9178 | 0.9495
> |    NeRF-SOS  | Unsupervised  | 0.9665 | 0.9912 | 0.9627 | 0.9769
> |    NVOS   | Weakly-supervised | 0.9217 | 0.9793 | 0.9145 | 0.9469
>
>
> **[Q4]:** The design choice of DINO-ViT and the contribution of DINO-ViT to the results?
> **[A4.1]:** DINO-ViT architecture is not applied by random picking, but with a strong motivation from previous literature. DINO-ViT firstly concludes that ViT architecture can extract stronger semantic information than ConvNets when being self-supervised trained. After that, many followers apply ViT architecture for the purpose of semantic guidance, including semantic appearance editing [7], unsupervised semantic segmentation [8], open-vocabulary semantic segmentation [9], and etc.
> **[A4.2]:** We could observe from the following experiments that only applies DINO to construct the “appearance-segmentation correlation” to distill DINO features into the segmentation field, achieves worse results than our full model on scene “Truck”. The results are consistent with the visualizations in Fig. 7 of the main draft, where we can see spatial discontinuities in the fourth column by using the DINO to construct the appearance correlation volume. It has been proven in [1] that the features from DINO-ViT may pay excessive attention to capturing semantic object parts rather than making the clusters spatial smooth. Therefore, our proposed geometric correlation volume is used to penalize discontinuities between neighboring points. The penalty is formulated by the attractive/repulsive force using point-wise distance via pulling/pushing positive/negative pairs.
>
> | Methods/Metrics |  IoU(BG) ↑  | IoU(FG) ↑ | mIoU ↑
> | :------------: |  :----: | :----: | :----:
> |   NeRF-SOS (App. only)    | 0.5029 | 0.4094 | 0.4562
> |   NeRF-SOS (Geo. only)  |  0.5516 | 0.4823 | 0.5169
> |   NeRF-SOS (Full Model)    | 0.9689 | 0.9455 | 0.9572
>
>
> **Reference:**
> [1]. Amir et al., Deep ViT Features as Dense Visual Descriptors
> [2]. sklearn: https://scikit-learn.org/stable/modules/generated/sklearn.cluster.KMeans.html
> [3]. Yu et al., Unsupervised Discovery of Object Radiance Fields
> [4]. Smith et al., Unsupervised discovery and composition of object light fields
> [5]. Ren et al., Neural Volumetric Object Selection
> [6]. Liu et al., Unsupervised Multi-View Object Segmentation Using Radiance Field Propagation
> [7]. Tumanyan et al., Splicing ViT Features for Semantic Appearance Transfer
> [8]. Hamilton et al., Unsupervised Semantic Segmentation by Distilling Feature Correspondences
> [9]. Xu et al., GroupViT: Semantic Segmentation Emerges from Text Supervision

---

> > ### Comment · Reviewer_vb78 · 2022-11-29
> > **Response continued...**
> >
> > Thank you authors for detailed explanations on all the points in the review. I am happy to find that you have made well-informed choices, and the additional experiments helped in clarifying the empirical questions. I am raising my score to 8.

---

> ### Author Response · Authors · 2022-11-14
> **Response to Reviewer vb78**
>
> **[Q5]:** Improve the writing clarity.
> **[A5]:** We have revised the writing in the updated draft for a clear technical details demonstration. Specifically, we clarify the limitation of previous methods, the advantage of DINO-based correspondence volume, and more description of the necessity of these techniques.
>
> **[Q6]:** Sparse labels for the training of Semantic-NeRF.
> **[A6]:** As Semantic-NeRF ables to perform label propagation with sparse annotation, we simulate sparse user annotation by randomly applying {1, 1%, 5%, 10%} foreground annotated object pixels while leaving the rest unlabeled. We can see in Fig. 9 of the appendix, the foreground boundaries are gradually refined when more annotations are included, which is consistent with the reported results in the original paper of Semantic-NeRF. However, as is shown in the following table, all sparse annotation experiments show insufficient accurate salient foreground segmentation, compared with densely annotated Semantic-NeRF. NeRF-SOS performs comparably with the dense annotation counterpart.
>
> | Methods/Metrics | User Click | NV-ARI ↑ | IoU(BG) ↑  | IoU(FG) ↑ | mIoU ↑
> | :------------: | :----:| :----: | :----: | :----: | :----:
> |   Semantic NeRF | 1 | 0.9615 | 0.9812 |  0.9528 | 0.967
> |   Semantic NeRF | 1% | 0.9731 | 0.9938 |  0.9667 | 0.9803
> |   Semantic NeRF | 5% | 0.9783 | 0.9950 |  0.9731 | 0.9841
> |   Semantic NeRF | 10% | 0.9788 | 0.9952 |  0.9735 | 0.9843
> |   Semantic NeRF | 100% | 0.9838 | 0.9963 |  0.9799 | 0.9881
> |    NeRF-SOS  | 0  | 0.9802 | 0.9955 | 0.9751 | 0.9853

---

### Official Review · Reviewer_9Jcm · 2022-10-25

**Confidence:** 3
**Correctness:** 4
**Technical Novelty And Significance:** 3
**Empirical Novelty And Significance:** 4
**Recommendation:** 8

**Clarity, Quality, Novelty And Reproducibility:**

The paper is well written but does not necessarily flow well.

The work appears of quality and technically sound.

The paper adapts the finding of Hamilton et al. (2022) into a NeRF framework, but provides a novel formulation for the geometry-level correspondence volume with a geometry-segmentation correlation formulation necessary for the approach to provide good results.

The authors provided the code in the supplementary material (although I don’t have compute resource available to test the reproducibility of the code)


**Strength And Weaknesses:**

Strengths:

The paper is well written and the method is technically sound.

The authors accommodated for a segmentation branch in the NeRF architecture, and the learning for that branch does not require any supervised training.

The results appear on par with fully-supervised approaches like Semantic-NeRF.

Weaknesses:

In the second phase of the training, it is unclear whether the color and density heads of the NeRF are frozen or not. Even though the \lambda_0 is set to 0, the other 2 losses can still induce a catastrophic forgetting of the knowledge learned in the 1st phase.If those are not frozen, the 2nd stage of the training might lead the appearance and geometry head of NeRF to collapse instead of distilling their knowledge into the segmentation head.

The cross-view geometric correspondence is unclear, in particular the motivation and insight on Equation (7), that appears to average the geometric coordinate t_k along the ray, based on the volumetric rendering based on the density learned by NeRF. Is that simply the point on the surface? Why not integrating some higher dimension features learned in NeRF? Also, for correctness, I believe \Delta_k in (7) should be \delta_k like in (1). Moreover, Equation (8) introduces spatial features g and g’ of dimension c, why is that the same dimension of f and f’ introduced in (3) and not of 3 dimension? Is the absolute distance an L2 norm?

In Equation (5) and (10), why \lambda_neg is not a negative value?


**Summary Of The Paper:**

The authors propose a self-supervised approach to segment objects from novel views synthesized from a NeRF.

The idea is borrowed from Hamilton et al. (2022) who learned semantic co-segmentation of images in an unsupervised fashion, by distilling the knowledge of DINO. The authors adapted the method to NeRF, to segment objects in novel views without any extra supervision. The knowledge is distilled from both visual and geometric features learned by NeRF.

The novel object segmentation branch in NeRF, learned by self-supervised distillation of the visual and geometric knowledge, show competitive performances with fully supervised methods such a Semantic NeRF, that require extra manual annotation of the masks.


**Summary Of The Review:**

The method provides very interesting insights on how to learn object segmentation in an unsupervised approach, leveraging consistency between multiple views in a NeRF representation.

The work appears technically sound, and paves the way for further work towards less supervised learning segmentation.

---

> ### Author Response · Authors · 2022-11-14
> **Response to Reviewer 9Jcm**
>
> **[Q1]:**  Whether the color and density heads of the NeRF are frozen or not?
> **[A1]:** To prevent the well-trained appearance and density branches from diverging, we fix the trained NeRF “backbone”, the appearance and density branches in the second phase. We have clarified this statement in the revised draft following your instruction.
>
> **[Q2]:** The motivation and insight of the cross-view geometric correspondence (a.k.a Eq. 7)?
> **[A2]:** The underlying motivation for introducing the “cross-view geometric correspondence” is to encourage spatial coherence of the distilled segmentation field, by formulating an attractive or repulsive force via the positive or negative pairs. For example, we can observe the clusters have spatial discontinuities when we only use appearance correspondence (see the 1st row, 4th column of Fig. 7 of the main draft). Such discontinuities are induced by the imprecise clusters, as the distilled features from DINO-ViT are excessively focused on capturing semantic object parts [1] rather than making the clusters spatial-coherent.
> Driven by this problem, we first leverage the “appearance-aware contrative loss” to make the distilled segmentation field cluster the scene features together if they are strong salient foreground signals. Then we further design a “geometry-aware contrastive loss” to make the distilled feature clusters spatial coherent. We have added the explanation in the revision.
>
> **[Q3]:** Is Eq.7 indicate the point on the surface? Why not integrate some higher-dimension features learned in NeRF?
> **[A3]:** Yes, it is the estimated depth value of that pixel. Following your suggestion, we switch the point-wise euclidean distance feature to the feature in the second last layer of the density field, to construct the geometric volume. As is shown in the following table, the two methods achieve similar segmentation accuracy on scene Fortress, validating the correctness and conciseness of the proposed point-wise geometric correlation volume.
>
> | Methods/Metrics | NV-ARI ↑ | IoU(BG) ↑  | IoU(FG) ↑ | mIoU ↑
> | :------------: | :----: | :----: | :----: | :----:
> |    NeRF-SOS (Feature-wise)   | 0.9789 | 0.9942 | 0.9725 | 0.9833
> |    NeRF-SOS (Point-wise, Ours)  | 0.9802 | 0.9955 | 0.9751 | 0.9853
>
> **[Q4]:** Typos
> **[A4]:** \Delta_k in Eq. 7 has the same meaning as \delta_k in Eq. 1, and we have corrected the typos, thanks! The notations of c in Eq. 3 and Eq. 8 indicate the DINO feature dimension and 3, respectively. We have added the statement to avoid confusion.
>
> **[Q5]:** Why \lambda_neg is not a negative value in Eq. 5 and Eq.10?
> **[A5]:** The sign of the loss value already reflects the attractive or repulsive force. Under the formulation of Eq.4 and Eq.9, minimizing Eq. 4 (with respect to S) encourages elements of S to be large, when F - b is positive. Whereas minimizing Eq. 4 (with respect to S) encourages elements of S to be small, when F - b is negative.
> Intuitively, we compute F and S using feature cosine similarities, positive pairs tend to generate positive F - b and exert attractive force. Negative pairs tend to generate negative F - b and exert repulsive force. Therefore, the weights of \lambda_id and \lambda_neg in Eq. 5 and Eq.10 are two positive values to balance the attractive/repulsive force.
>
> **Reference:**
> [1]. Amir et al., Deep ViT Features as Dense Visual Descriptors

---

### Official Review · Reviewer_LCEX · 2022-10-25

**Confidence:** 3
**Correctness:** 3
**Technical Novelty And Significance:** 3
**Empirical Novelty And Significance:** 3
**Recommendation:** 8

**Clarity, Quality, Novelty And Reproducibility:**

Good. The paper is well-written. The authors describe their method in detail, which guides for reproducing the result.
The authors provide various results on different datasets and the comparison of different methods, making the results convincible.
The proposed method is somewhat innovative and gets more optimal segmentation results.


**Strength And Weaknesses:**

Strengths:
1. This is a well-written paper.
2. The proposed method is compared with various methods. The experiments are complete and convincing.
3. Some visualizations are helpful to understand.
Weaknesses:
1. Although the visual results of the proposed method are better than the supervised method. The evaluation metrics of the proposed method are worse than Semantic-NeRF in Tab 2 and Tab 3. What causes that problem?
2. The paper lacks a quantitative analysis of the effects. May you give an analysis of your design?
3. The ablations are not sufficient enough. Ablation should contain all of your extra design including geometry contrastive loss and appearance contrastive loss.

**Summary Of The Paper:**

This work proposes an appearance contrastive loss to apply the self-supervised learned 2D visual feature for 3D representations.
A new geometry contrastive loss is proposed for object segmentation to involve geometric information for segmentation clustering.
The segmentation of the proposed method has a more refined result than its supervised counterpart.

**Summary Of The Review:**

The paper is well-written, and the result of object segmentation is better than its counterpart obviously.
I think it is marginally above the acceptance threshold.

---

> ### Author Response · Authors · 2022-11-14
> **Response to Reviewer LCEX**
>
> **[Q1]:** The evaluation metrics of the proposed method are worse than Semantic-NeRF, although the visual results of the proposed method are better.
>
> **[A1]:** The reason is: Semantic-NeRF fits the annotations while NeRF-SOS is blind to these annotations. However, pixel-wise annotation on 2D images is difficult, especially for objects with holes [1]. We leverage polygons [2] to create the contour of foreground objects, following the annotation convention of the representative panoptic segmentation dataset [3]. Therefore, the annotations' accuracy may be limited by the number of edges in these polygons. We have tried to annotate more than 50 edges for each object on each view to guarantee the annotation quality. Moreover, to avoid any unfair comparisons, we provide all the predicted masks used for evaluation in our appendix and supplementary video.
>
> **[Q2]:** Quantitative analysis of the designed geometry contrastive loss and appearance contrastive loss.
>
> **[A2.1]:** The effectiveness of the proposed two-level collaborative contrastive loss can be verified in the following table. As can be seen in the first and second rows, using only appearance or geometric contrastive loss leads to inferior segmentation results on scene Truck. The underlying reasons are: clustering based only on appearance features may lead to spatial discontinuities (see the 1st row, 4th column of Fig. 7 in the main draft) when we only use appearance correspondence. Such discontinuities are induced by the distilled features from DINO-ViT are overfocus capturing semantic object parts [4] rather than making the clusters spatial coherent. Therefore, we further design a “geometry-aware contrastive loss” to make the distilled feature clusters spatial coherent, by formulating attractive/repulsive force using positive/negative pairs based on scene geometry.
>
> | Methods/Metrics |  IoU(BG) ↑  | IoU(FG) ↑ | mIoU ↑
> | :------------: |  :----: | :----: | :----:
> |   NeRF-SOS (App. only)    | 0.5029 | 0.4094 | 0.4562
> |   NeRF-SOS (Geo. only)  |  0.5516 | 0.4823 | 0.5169
> |   NeRF-SOS (Full Model)    | 0.9689 | 0.9455 | 0.9572
>
> **[A2.2]:** Besides, our design does not require any change to the core NeRF model architectures, by only appending a parallel segmentation branch and distilling appearance/geometric levels feature using a collaborative contrastive loss in the second phase. We have validated this point by implementing the collaborative loss upon NeRF and NeRF++.
>
> **Reference:**
> [1]. Pavoni et al., TagLab: AI-assisted annotation for the fast and accurate semantic segmentation of coral reef orthoimages
> [2]. labelme: https://github.com/wkentaro/labelme
> [3]. Mapillary Vistas Dataset. https://www.mapillary.com/dataset/vistas
> [4]. Amir et al., Deep ViT Features as Dense Visual Descriptors

---

> ### Comment · Reviewer_LCEX · 2022-11-17
> **Brief summary**
>
> The authors have addressed all my concerns. I have no other comments.

---

### Author Response · Authors · 2022-11-14
**General Response**

We thank all reviewers for their insightful and constructive suggestions. We are glad that reviewers found: (1). The work is technically sound (Reviewer 9Jcm), novel and effective (viewer vb78); (2). The experiments are complete and convincing (Reviewer LCEX), and show competitive/good performances with fully-supervised method (Reviewer 9Jcm and 92aZ); (3). The paper is well-written (Reviewer LCEX, 9Jcm).

We have addressed all the questions that the reviewers posed with additional experimental results. We have carefully modified our manuscript, following those suggestions.

---

### Decision · Program_Chairs · 2023-01-20

**Decision:**

Accept: poster

**Justification For Why Not Higher Score:**

This paper falls into a large number of NeRF for visual recognition papers. Compared with prior or concurrent work (e.g., Unsupervised Multi-View Object Segmentation Using Radiance Field Propagation from NeurIPS 2022), this paper does not stand out.

**Justification For Why Not Lower Score:**

Reviewers are all positive about the submission.

**Metareview: Summary, Strengths And Weaknesses:**

This submission explores the use of NeRF for image segmentation in a self-supervised way. Reviewers overall appreciate the extensive experiments and the model design. During the rebuttal, the authors were able to address the concerns about missing related work and presentation. The AC agrees with the reviewers and recommends acceptance. The authors should include the additional discussion and results in the camera ready.

**Note From Pc:**

if the above contains the word "oral" or "spotlight" please see: "oral" presentation means -> notable-top-5% and "spotlight" means -> notable-top-25%. As stated in our emails, we are disassociating presentation type from AC recommendations